# Normalizing Kalman Filters for Multivariate Time Series Analysis

**Emmanuel de Bézenac**[1][†][∗] **Syama Sundar Rangapuram**[2][∗]**, Konstantinos Benidis**[2]**,**
**Michael Bohlke-Schneider**[2]**, Richard Kurle**[3][‡] **Lorenzo Stella**[2]**,**
**Hilaf Hasson**[2]**, Patrick Gallinari**[1]**, Tim Januschowski**[2]
[1]Sorbonne Université, [2]AWS AI Labs, [3]Technical University of Munich

Correspondence to: `emmanuel.de-bezenac@lip6.fr`, `rangapur@amazon.de`

## Abstract

This paper tackles the modelling of large, complex and multivariate time series panels in a probabilistic setting. To this extent, we present a novel approach reconciling classical state space models with deep learning methods. By *augmenting* state space models with normalizing flows, we mitigate imprecisions stemming from idealized assumptions in state space models. The resulting model is highly flexible while still retaining many of the attractive properties of state space models, e.g., uncertainty and observation errors are properly accounted for, inference is tractable, sampling is efficient and good generalization performance is observed, even in low data regimes. We demonstrate competitiveness against state-of-the-art deep learning methods on the tasks of forecasting real world data and handling varying levels of missing data.

## 1   Introduction

In most real world applications of time series analysis, e.g., risk management in finance, cannibalization of products in retail or anomaly detection in cloud computing environments, time series are not mutually independent and an accurate modelling approach must take these dependencies into account [1]. The classical approach [2] is to extend standard univariate models resulting in vector autoregression [3], multivariate GARCH [4] and multivariate state space models [5, 6]. Although these approaches yield useful theoretical properties, they make idealized assumptions like Gaussianity, linear inter-dependencies, and are not scalable to even moderate number of time series [7] due to the number of parameters required to be estimated, which is restrictive for many modern applications involving large panels of time series. Recently, more expressive, scalable deep learning methods [8, 9] were developed for forecasting applications that learn a joint global model for multiple time series; however, they still assume that these time series are mutually independent.

In this paper we propose the *Normalizing Kalman Filter* (NKF), a novel approach for modelling and forecasting complex multivariate time series by augmenting classical linear Gaussian state space models (LGM) with normalizing flows [10]. The combined model allows us to leverage the flexibility of normalizing flows (NF), alleviating strong assumptions of traditional multivariate models, while still benefiting from the rich set of mathematical properties of LGM. In fact, we prove that despite modelling non-Gaussian data with nonlinear inter-dependencies, we can achieve exact inference since our model has closed-form expressions for filtering, smoothing and likelihood computation. We thus retain the main attractive properties of LGM, in contrast to related methods [11, 12]. Moreover, since

---

[∗]Equal contribution.
[†]Work done while at Amazon.

our model is based on LGM, handling of missing data and integrating prior knowledge, e.g., seasonality and trend, becomes trivial. Therefore, the proposed model can be used in forecasting time series with missing or noisy data irrespective of whether the data regime is sparse (in terms of observed time points) or dense. More importantly, LGM directly gives us the ability to provide tractable multi-step ahead forecast distributions while accounting for all uncertainties; this is in contrast to recent deep learning-based autoregressive models [8, 1] that do not incorporate accumulated prediction errors into forecast distributions since predictions of the model are used as lagged inputs in a multi-step forecast scenario. For the forecasting application, we show that our method scales linearly with the number of dimensions and number of time points, unlike most of the existing work that exhibits quadratic scaling with the number of dimensions. The necessary structural assumptions do not result in a loss of generality or expressiveness of the overall model.

In summary, our main contributions are as follows:

○ A tractable method for modelling non-Gaussian multivariate time series data with nonlinear inter-dependencies that has Kalman-like recursive updates for filtering and smoothing.
○ A scalable, robust multivariate forecasting method that handles missing data naturally and provides tractable multi-step ahead forecast distributions while accounting for uncertainties unlike autoregressive models [8, 1].
○ A thorough evaluation of applicability of normalizing flows in the context of high-dimensional time series forecasting for handling non-Gaussian multivariate data with nonlinear dependencies.

## 2   Normalizing Kalman Filters

Let $y_t \in \mathbb{R}^N$ denote the value of a multivariate time series at time $t$, with $y_{t,i} \in \mathbb{R}$ the value of the corresponding $i$-th univariate time series. Further, let $x_{t,i} \in \mathbb{R}^k$ be time varying covariate vectors associated to each univariate time series at time $t$, and $x_t := [x_{t,1}, \ldots, x_{t,N}] \in \mathbb{R}^{k \times N}$. Non-random and random variables are denoted by normal and bold letters, i.e., $x$ and $\mathbf{x}$, respectively. We use the shorthand $y_{1:T}$ to denote the sequence $\{y_1, y_2, \ldots, y_T\}$.

### 2.1   Generative Model

The core assumption behind our Normalizing Kalman Filter (NKF) model is the existence of a latent state that evolves according to simple (linear) dynamics, with potentially complex and nonlinear dependencies between latent state and observations–and thus, among observations. More precisely, the dynamics of the latent state $\mathbf{l}_t \in \mathbb{R}^d$ are governed by a time-dependent *transition matrix* $F_t \in \mathbb{R}^{d \times d}$, up to additive Gaussian noise $\boldsymbol{\epsilon}_t$ as in (1b). The state is then mapped into the space of observations with *emission matrix* $A_t \in \mathbb{R}^{d \times N}$, and additive Gaussian noise $\boldsymbol{\varepsilon}_t$ before being transformed by a potentially nonlinear function $f_t : \mathbb{R}^N \to \mathbb{R}^N$ parametrized by $\Lambda$, generating observation $\mathbf{y}_t \in \mathbb{R}^N$:

$$
\begin{aligned}
& \mathbf{l}_1 \sim \mathcal{N}(\mu_1, \Sigma_1) && \text{(initial state)} && \text{(1a)} \\
\text{(NKF model)} \quad & \mathbf{l}_t = F_t \mathbf{l}_{t-1} + \boldsymbol{\epsilon}_t, && \boldsymbol{\epsilon}_t \sim \mathcal{N}(0, \Sigma_t), && \text{(transition dynamics)} && \text{(1b)} \\
& \mathbf{y}_t = f_t(A_t^T \mathbf{l}_t + \boldsymbol{\varepsilon}_t), && \boldsymbol{\varepsilon}_t \sim \mathcal{N}(0, \Gamma_t). && \text{(observation model)} && \text{(1c)}
\end{aligned}
$$

With parameters $\Lambda$ and $\Theta = (\mu_1, \Sigma_1, \{\Gamma_t, A_t\}_{t \geq 1}, \{\Sigma_t, F_t\}_{t \geq 2})$, the model is fully specified.[3] Note that the special case $f_t = \text{id}$ recovers the standard LGM where both the transition dynamics and the observation model are linear. In the following, this similarity with LGM will yield numerous computational benefits and it will further allow us to easily inject prior knowledge on the structural form of the dynamics (e.g., levels, trends, seasonalities [5]) for good generalization properties.

We consider a flexible nonlinear transformation for the observation model, assuming invertibility of $f_t$. This guarantees the conservation of probability mass and allows the evaluation of the associated density function at any given point of interest. In particular, the probability density of an observation

$y_t$ given the state $l_t$ can be computed using the change of variables formula:

$$p(y_t|l_t; \Theta, \Lambda) = p_z(f_t^{-1}(y_t)|l_t; \Theta) \left| \det \left[ \text{Jac}_{y_t}(f_t^{-1}) \right] \right|, \tag{2}$$

where the first term in $p_z(z_t|l_t; \Theta)$ is the density of the Gaussian variable $z_t := A_t l_t + \varepsilon_t$ conditioned on $l_t$, and the second is the absolute value of the determinant of the Jacobian of $f_t^{-1}$ given $\Lambda$, evaluated at $y_t$. This equation and Figure 1 illustrate the intuition behind our approach: we would like $f_t^{-1}$ to transform the observations such that the dynamics become simple and the noise is Gaussian.

Computing the density (2) raises several issues: (i) finding a flexible $f_t$ while ensuring invertibility, (ii) being able to compute the inverse efficiently and (iii) tractability of the computation of the Jacobian term when the number of time series $N$ is large. To this extent, we will take inspiration from *normalizing flows* [13, 14, 15], which are invertible neural networks that typically transform isotropic Gaussians to fit a more complex data distribution.

These invertible networks are tailored to compute the Jacobian term efficiently. Moreover, they have proven to work very well for nonlinear high-dimensional data, e.g., images [14], both in terms of flexibility and generalization. In our approach, we apply these invertible neural networks to temporal data, using them to map the distribution $p_z$ given by the LGM to the complex data distribution. This yields a powerful function $f_t$ where the Jacobian term is computable in *linear time* in $N$.

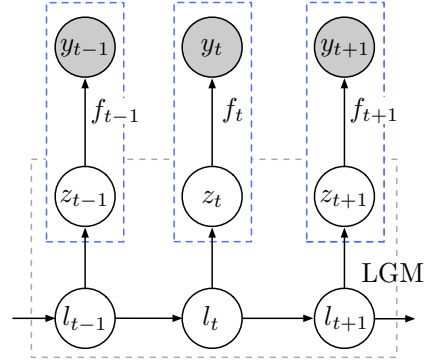

Figure 1: Generative model of the NKF. States $l_t$ and *pseudo-observations* $z_t$ are produced from an LGM, which are then transformed through a normalizing flow $f_t$, producing observations $y_t$.

**Inference and Learning.** With the presented generative model it is possible to do inference (e.g., filtering and smoothing) and training in a simple and tractable way. Similar to the LGM, computing the *filtered distribution* $p(l_t|y_{1:t}; \Theta, \Lambda)$ is essential as it determines our current belief on the state having observed all the data up to time $t$, and it takes part in the computation of the likelihood of the model parameters as well as the forecast distribution. For general nonlinear state space models its computation is tedious as it involves integrating out previous states.

Methods such as Particle Filters [16] resort to Monte Carlo approximations of these integrals but have difficulty scaling to high dimensions. Other methods circumvent this by locally linearizing the nonlinear transformation [17] or by using a finite sample approximation [18] in order to apply–in both cases– the techniques of the standard LGM, but introduce a bias. In contrast, our model allows for computing this quantity in a tractable and efficient manner, without resorting to any form of simplification or approximation. In fact, despite the nonlinear nature of $f_t$, the filtered distribution *remains* Gaussian and its parameters can be computed in closed form similarly to the Kalman Filter, as shown in the following proposition.

**Proposition 1** (Filtering). The *filtered* distributions of the NKF model are *Gaussian* and are given by the filtered distributions of the corresponding LGM with pseudo-observations $z_t := f_t^{-1}(y_t)$, $t \geq 1$. That is, $p(l_t|y_{1:t}; \Theta, \Lambda) = p_{\text{LGM}}(l_t|z_{1:t}; \Theta)$ where $p_{\text{LGM}}$ refers to the distribution given by the LGM.

This can be proved by induction using recursive Bayesian estimation and is available in Appendix A.1 along with the exact updates. Proposition 1 shows that filtered distributions for our model are available in closed-form and have the same computational complexity as that of the LGM, plus the complexity of the inverse of the nonlinear function $f$ and the Jacobian term in (2).[4]

Our nonlinear model (1) is also amenable to smoothing, i.e., computing the smoothed posterior distribution $p(l_t|y_{1:T}; \Theta, \Lambda)$, given past, present and future observations. Smoothing is a prevalent problem in many fields, with applications such as estimating the distribution of missing data and providing explanations in the context of offline anomaly detection. Smoothing can be obtained with a backward iterative approach using the quantities computed during a preliminary filtering pass, starting from the filtered distribution corresponding to step $T$, $p(l_T|y_{1:T}; \Theta)$. Similar to filtering, smoothing updates also directly translate to the corresponding updates of the LGM.

**Proposition 2** (Smoothing). The *smoothed* distributions of the NKF model are *Gaussian* and are given by the smoothed distributions of the corresponding LGM with pseudo-observations $z_t := f_t^{-1}(y_t), t = 1, 2 \ldots, T$. That is, $p(l_t|y_{1:T}; \Theta, \Lambda) = p_{\text{LGM}}(l_t|z_{1:T}; \Theta)$.

The tractability of the computation of the filtered distribution implies that the likelihood of the model parameters can be computed efficiently by integrating out the latent state. In case of the standard LGM, the likelihood of its parameters $\Theta$ given the observations $z_{1:T}$ can be written as

$$\ell(\Theta) = \prod_{t=1}^{T} p_{\text{LGM}}(z_t|z_{1:t-1}; \Theta), \tag{3}$$

where $p_{\text{LGM}}(z_t|z_{1:t-1}; \Theta)$ is the predictive distribution of LGM. This distribution is available in closed-form thanks to the filtering updates [19]. We now show that the likelihood of our nonlinear model given the observations $\{y_{1:T}\}$ is essentially a reweighted version of this expression with $z_t = f_t^{-1}(y_t)$.

**Proposition 3** (Likelihood). The likelihood of the parameters $(\Theta, \Lambda)$ of the NKF model given the observations $\{y_{1:T}\}$ can be computed as

$$\ell(\Theta, \Lambda) = p(y_{1:T}; \Theta, \Lambda) = \prod_{t=1}^{T} p_{\text{LGM}}(z_t|z_{1:t-1}; \Theta) \left| \det \left[ \text{Jac}_{z_t}(f_t) \right] \right|^{-1}, \tag{4}$$

where $z_t = f_t^{-1}(y_t)$ and $p_{\text{LGM}}(z_t|z_{1:t-1}; \Theta)$ denotes the predictive distribution of LGM.

Hence, we can compute the likelihood of the parameters of the NKF exactly (Appendix A.3 contains the closed-form expressions), and fit our model to the given data by directly maximizing the likelihood. This is done without resorting to any approximations typically required in other nonlinear extensions of LGM such as the Particle Filter and the Extended/Unscented Kalman Filter.

## 2.2 Parameter Estimation using RNNs

In Sec. 2.1, we assumed the data was generated from a given family of NKF state space models characterized by its parameters $\Theta_{1:T} = (\mu_1, \Sigma_1, \{\Gamma_t, A_t\}_{t=1}^T, \{\Sigma_t, F_t\}_{t=2}^T)$ along with $\Lambda$, the parameters of the normalizing flow transformation $f_t$ (which we assume to be constant over time). However the exact values of the parameters are unknown. Similar to [9, 8], we propose to predict the parameters $\Theta$ from the covariates, using a recurrent neural network (RNN) where the recurrent function $\Psi$ is parametrized by $\Phi$, taking into account the possibly nonlinear relationship between covariates $x_t$:

$$\Theta_t = \sigma(h_t; \Phi), \quad h_t = \Psi(x_t, h_{t-1}; \Phi), \quad t = 1, \ldots T, \tag{5}$$

where $\sigma$ denotes the transformation mapping of the RNN output to domains of the parameters (Appendix B.1). The RNN allows our model to be more expressive by allowing time-varying parameters along with temporal dependencies in the covariates, a requirement for forecasting. The parameters of the RNN $\Phi$ and of the normalizing flow $\Lambda$ are estimated by maximizing the conditional likelihood,

$$p(y_{1:T}|x_{1:T}; \Phi, \Lambda) := p(y_{1:T}; \Theta_{1:T}, \Lambda) \tag{6}$$

where $p(y_{1:T}; \Theta_{1:T}, \Lambda)$ is given by (4). Although the transition model in our case is linear, complex dynamics can still be taken into account as the parameters of the LGM, which are now the outputs of a RNN, may change in a nonlinear way.

## 2.3 Applications: Forecasting and Missing Values

Given a model for sequential data, we can apply it to obtain future time steps $T+1:T+\tau$, or in other words, *forecasts* of the time series, starting from past observations $y_{1:T}$ and covariates $x_{1:T+\tau}$.

In our case such a forecast distribution $p(y_{T+1:T+\tau}|y_{1:T}, x_{1:T+\tau}; \Phi, \Lambda)$ can also be computed efficiently and is available in closed-form. From this, not only can we readily evaluate possible future scenarios (see Appendix A.4), but also draw samples from it to generate forecasts.

This is achieved by first computing the filtered distribution for the input range $1 : T$, and then recursively apply the transition equation and the observation model to generate prediction samples. More precisely, starting with the RNN state $h_T$ and $l_T$ sampled from $p(l_T|y_{1:T}, x_{1:T}; \Theta_{1:T})$, given by (5) and (6), respectively, we iteratively apply:

$$F_t, A_t, \Sigma_t, \Gamma_t = \sigma(h_t; \Phi), \qquad h_t = \Psi(x_t, h_{t-1}; \Phi), \qquad (7a)$$

$$l_t = F_t l_{t-1} + \epsilon_t, \qquad \epsilon_t \text{ sampled from } \mathcal{N}(0, \Sigma_t), \qquad (7b)$$

$$y_t = f_t(A_t^T l_t + \varepsilon_t), \qquad \varepsilon_t \text{ sampled from } \mathcal{N}(0, \Gamma_t), \qquad t = T+1, \dots, T+\tau. \qquad (7c)$$

In contrast to alternative deep learning approaches [1, 8, 11], this generative procedure is *not* autoregressive in the sense that observations $y_t$ are never fed to the model. Instead, it is the filtering that correctly updates our beliefs based on the observed data. This has several notable consequences: we do not have to resort to large and cumbersome beam searches to obtain proper uncertainty estimates, noisy observations with varying levels of uncertainty can be properly handled, and long-term forecast computations are accelerated as intermediate observations need not be computed.

In many real world applications, observations for each time step may not be available, e.g., out-of-stock situations in the context of demand time series (no sales does not mean no demand if out-of-stock) or network failures in the case of sensor data streams. *Handling missing data* is then of central importance and there are two aspects to it: (i) learning in the presence of missing values without imputing them and (ii) imputing the missing values. Similar to LGM [20], our approach offers a straightforward, tractable and unbiased way for handling missing data in both of these scenarios.

In case of learning with missing values the likelihood terms corresponding to the missing entries should be ignored. This amounts to effectively dealing with them in the filtering step. Without loss of generality, let us assume that targets until time $t-1$ are observed but are missing for $t$. In this case, we can compute the filtered distribution $p(l_{t-1}|y_{1:t-1}; \Theta)$ with the method outlined in Sec. 2. As $y_t$ is not observed, the filtered distribution at time $t$ then simply corresponds to $p(l_t|y_{1:t-1}; \Theta)$, and can be obtained by applying the prediction step, starting from the filtered distribution at time $t-1$.

For the second case, we wish to impute missing values at time $t \notin \mathbf{t}^{\text{obs}}$ based on observed values at times $\mathbf{t}^{\text{obs}}$. This can be easily achieved by computing the smoothed distribution $p(l_t|y_{\mathbf{t}^{\text{obs}}}; \Theta)$ in the presence of missing data, and from this compute $p(y_t|y_{\mathbf{t}^{\text{obs}}}; \Theta)$ (full details in Appendix A.5). This distribution is available in closed form and can be readily sampled from. This allows us to use the same model for forecasting and imputation without the need for bidirectional RNNs [21] as the smoothing procedure inherently handles future data.

## 3 Local-Global Instantiation

The number of parameters of the NKF scales quadratically in the dimensionality $d$ of the state (stemming from the covariance matrices of the Gaussian distributions) in the worst case. Leveraging the strength of NF, we present an instantiation of NKF with an induced *local-global* structure, that displays several advantages over the general unstructured form, e.g., exhibiting linear scaling in $d$.

We assume that each time series is associated with latent factors that evolve *independently w.r.t.* the factors of the other time series. These factors will in turn be *mixed* together with the normalizing flow, producing *dependent* time series observations. Formally, for each univariate time series $i$ we associate a *local* LGM with parameters $\Theta^{(i)} = (\mu_1^{(i)}, \Sigma_1^{(i)}, \{\Gamma_t^{(i)}, A_t^{(i)}\}_{t \geq 1}, \{\Sigma_t^{(i)}, F_t^{(i)}\}_{t \geq 2})$, whose dynamics are characterized by:

$$\mathbf{l}_t^{(i)} = F_t^{(i)} \mathbf{l}_{t-1}^{(i)} + \boldsymbol{\epsilon}_t^{(i)}, \quad \boldsymbol{\epsilon}_t^{(i)} \sim \mathcal{N}(0, \Sigma_t^{(i)}), \qquad (8a)$$

$$\mathbf{z}_t^{(i)} = A_t^{(i)T} \mathbf{l}_t^{(i)} + \boldsymbol{\varepsilon}_t^{(i)}, \quad \boldsymbol{\varepsilon}_t^{(i)} \sim \mathcal{N}(0, \Gamma_t^{(i)}), \qquad (8b)$$

$$\mathbf{y}_t = f_t(\mathbf{z}_t^{(1)}, \dots, \mathbf{z}_t^{(N)}). \qquad (8c)$$

Note that here $\Gamma_t^{(i)}$ is a scalar and denotes variance of the Gaussian noise. This local-global structure has several advantages: (i) the dynamics of the local states need not be the same for each time series and thus, prior knowledge on the evolution can be readily injected *per* time series, (ii) computations can be done in parallel for each time series and finally (iii) we may benefit from the effects of amortization by predicting local parameters for each time series and sharing the same weights $\Phi$:

$\Theta_t^{(i)} = \Psi(x_t^{(i)}, h_{t-1}^{(i)}; \Phi)$, allowing the RNN to make analogies between time series. In this case, dependencies across time series are captured with the normalizing flow $f_t$, mixing the components together in a nonlinear fashion.

We now explain a possible instantiation of the LGM (Eq. (8a), (8b)), which is an innovation-based model, similar to [9]. Note that this is a choice and not a requirement as any other LGM instance may be used if that better reflects the data at hand. Nonetheless, with this form a wide range of phenomena can be captured, e.g., long term direction (trends), patterns that repeat (seasonality), cycles with different periodicity, multiplicative errors, etc.; we refer the reader to [5]. In our instantiation, we combine both level-trend and seasonality components, as described in Appendix B.2.

For $f_t$, we use the RealNVP architecture [13]: it is a network that is composed of a series of parametrized invertible transformations with a lower triangular Jacobian structure and vector component permutations in order to capture complex dependencies. This structure has the advantage of being flexible, while maintaining a tractable Jacobian term, computable in linear time. Moreover, as opposed to more recent architectures e.g., [14], the number of parameters scales linearly. For the RNN, we use an LSTM with 2 layers.

In terms of parameter complexity, this particular instantiation scales as $O(N)$ for each timestep, due to the diagonal structure of the covariance matrices (although they do not correspond to the effective number of parameters as these are predicted from the RNN). Moreover, time complexity for likelihood computation and forecasting scales as $O(N(d^2 + k))$ for each timestep, where $k$ is the number of covariates and $d$ is the dimension of latent state; in experiments $d = 32$ (24 hourly components + 7 daily components + one level component) for hourly data and $d = 8$ for daily data.

**Partial observations:** The local-global instantiation model presented above could deal with partial observations (i.e., only some entries of $y_t \in \mathbb{R}^N$ are missing but not all); however it would require marginalisation over the missing dimensions, which cannot be done in closed-form. Alternatively, if one is interested in dealing with partial observations, it is possible to consider an instantiation with a global (i.e., multivariate) LGM with non-diagonal covariance matrix modelling *linear* dependencies among the time-series $y_t \in \mathbb{R}^N$ at any given time step $t$, and a normalising flow applied locally to each time-series: $y_t = f_t(z_t) = (u_t(z_{1,t}), \ldots, u_t(z_{N,t}))$, with $u_t : \mathbb{R} \to \mathbb{R}$ (see ablation study in our experiments). In this case, the marginalisation of any missing set of time series actually yields an analytic form for the filtering, smoothing and forecast distribution [20], and hence the handling of partial observations can be efficiently dealt with.

## 4 Experiments

### 4.1 Qualitative Results

We qualitatively assess our approach in Section 3 with two synthetic datasets S1 and S2 with increasing difficultly. The datasets are composed of 2 daily univariate time series with a weekly seasonality pattern. We compare our approach against a variant without any normalizing flow (this amounts to setting $f_t = $ id in (8)). For dataset S1 the daily data has three different modes and highly non-Gaussian time-*independent* observational noise. S2 is similar, but the time series are *mixed* together in a nonlinear fashion, and the observational noise is not only non-Gaussian, but time-*dependent* (Appendix C.1 contains a detailed description). Target data (blue) and forecasts (green) are plotted in Fig. 2, where the first and second axis correspond to the first and second time series respectively, and data is aggregated over the temporal axis.

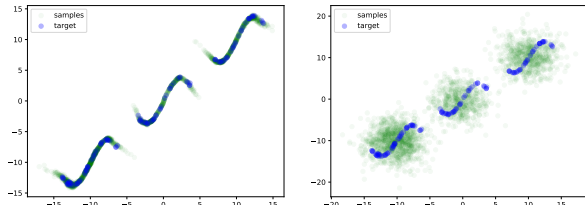

(a) S1 : Results with (left) and without (right) NF.

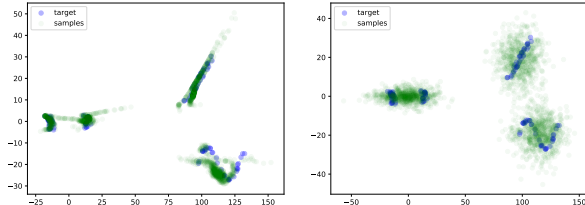

(b) S2 : Results with (left) and without (right) NF.

Figure 2: Evaluation w/ and w/o NF. The axes correspond to the two components of the 2D time series; the temporal axis is marginalized out. Green points correspond to model samples when predicting 24 days ahead and blue points to future samples from the data-generating process.

| Approach | VES | VAR | GARCH | DeepAR | GP-Copula | KVAE | NKF(Ours) |
|---|---|---|---|---|---|---|---|
| Multivariate | ✓ | ✓ | ✓ | × | ✓ | ✓ | ✓ |
| Non-Linear, Non-Gaussian | × | × | × | ✓ | ✓ | ✓ | ✓ |
| Filtering & Smoothing | ✓ | NA | NA | NA | NA | ✓ | ✓ |
| Tractable Multi-step Forecast | ✓ | × | × | × | × | × | ✓ |
| Tractable Data Imputation | ✓ | × | × | × | × | × | ✓ |

Table 1: Comparative summary of competing approaches on various parameters.

Observations and forecasts can also be seen in Appendix C.1 from a viewpoint that better highlights the seasonal nature of the observations.

For both datasets, the variant with $f_t = \mathtt{id}$ captures the daily modes correctly, but assumptions such as Gaussianity and independence between time series introduce errors. In contrast, visual inspection reveals that applying the normalizing flow allows us to fit the data better, capturing the complex dependencies between time series and the non-Gaussian noise, for both S1 and S2.

## 4.2 Quantitative Results

We follow the experimental set up proposed in [1] since it focusses on the same problem in the forecasting application. In particular, we evaluate on the public datasets used in [1]; see C.2.1 for details and Table 3 for the summary of datasets. Similar to [1], the forecasts of different methods are evaluated by splitting each dataset in the following fashion: all data prior to a fixed *forecast start date* compose the training set and the remainder is used as the test set. We measure the accuracy of the forecasts of various methods on all the time points of the test set. The hyperparameters have been selected using a validation set of equal size to the test set, created from the last time steps of the training data.

Our evaluation is extensive, covering relevant classical multivariate approaches as well as recent deep learning based models. In particular, we compare against VES, a direct generalization of univariate innovation state space model (a special case of LGM used in forecasting) to multivariate time series (see Chapter 17 of [5]), VAR, a multivariate linear autoregressive model and GARCH a multivariate conditional heteroskedastic model [22]; we include result of Lasso-regularized VAR as well. We also compare against the recent deep-learning based approaches GP-Copula [1] and KVAE [12] specifically developed for handling non-Gaussian multivariate data with non-linear dependencies. GP-Copula builds on ideas of VAR and relies on RNN and low-rank Gaussian Copula process for going beyond Gaussianity and linearity. In contrast, KVAE uses a variational autoencoder on top of linear state space models to achieve the same. Unlike our NKF model, inference and likelihood computation are not tractable in KVAE and it relies on particle filters for their approximation. Additionally, we compare with DeepAR [8] an autoregressive recurrent neural network based method for univariate time series forecasting. DeepState [9] is a special case of NKF model and is part of the ablation study. See Table 1 for summary of the compared methods based on various parameters.

In order to evaluate forecasting models, *continuous ranked probability score* (CRPS) is generally accepted as one of the most well-founded metrics[23, 24]. However, this metric is only defined for univariate timeseries and cannot assess if dependencies across time series are accurately captured. Different generalizations to the multivariate case have been used, e.g., the energy score, or CRPS-Sum [1]. We have opted for the CRPS-Sum, as the energy score suffers from the curse of dimensionality, from both a statistical and computational viewpoint [25, 26]. The introduction of the CRPS-Sum [1] was experimentally justified. In this work, we prove it is theoretically sound as justified formally in Appendix C.4 following the proper scoring rule framework [24]. Note that, opposed to [1], as different time series observations may have drastically different scales, we first normalize each time series by the sum of its absolute values before computing this metric (hence the '-N' suffix). We also did not choose log-likelihood since not all methods yield analytical forecast distributions and is not meaningful for some methods [1].

We report CRPS-Sum-N metric values achieved by all methods in Table 2. Classical methods, because of the Gaussianity and linear dependency assumptions, typically yield inferior results; entries marked with '-' are runs failed with numerical issues. Deep learning based models have superior

| method | exchange | solar | elec | wiki | traffic |
|---|---|---|---|---|---|
| VES | **0.005 ± 0.000** | 0.9 ± 0.003 | 0.88 ± 0.0035 | - | 0.35 ± 0.0023 |
| VAR | **0.005 ± 0.000** | 0.83 ± 0.006 | 0.039 ± 0.0005 | - | 0.29 ± 0.005 |
| VAR-Lasso | 0.012 ± 0.0002 | 0.51 ± 0.006 | 0.025 ± 0.0002 | 3.1 ± 0.004 | 0.15 ± 0.002 |
| GARCH | 0.023 ± 0.000 | 0.88 ± 0.002 | 0.19 ± 0.001 | - | 0.37 ± 0.0016 |
| DeepAR | 0.006±0.001 | 0.336±0.014 | 0.023±0.001 | 0.127±0.042 | 0.055±0.003 |
| GP-Copula | 0.007±0.000 | 0.363±0.002 | 0.024±0.000 | 0.092±0.012 | **0.051±0.000** |
| KVAE | 0.014 ± 0.002 | 0.34 ± 0.025 | 0.051 ± 0.019 | 0.095 ± 0.012 | 0.1 ± 0.005 |
| NKF(Ours) | **0.005 ± 0.000** | **0.320±0.020** | **0.016±0.001** | **0.071±0.002** | 0.10±0.002 |
| ablation study $f_t = $ id | 0.005±0.000 | 0.415±0.002 | 0.026±0.000 | 0.082±0.000 | 0.123±0.000 |
| $f_t$ Local | 0.005±0.000 | 0.405±0.005 | 0.018±0.001 | 0.068±0.004 | 0.102±0.013 |

Table 2: CRPS-Sum-N (lower is better), averaged over 3 runs. The case $f_t = $ id is `DeepState` [9] and VES can be seen as part of ablation where normalizing flow and RNN are removed from `NKF`.

performance overall. In particular, `NKF` achieves the best result in 4 out of 5 datasets. On `traffic` `NKF` is better than all methods except for `DeepAR` and `GP-Copula` which are purely data-driven autoregressive approaches with minimal modelling assumptions. Given the domain of `traffic` dataset is $(0, 1)$, it would be interesting to verify in future work if the relatively weak performance is due to a short-coming of the normalizing flow part or due to the modelling choice of adopting a state space model instead of an autoregressive process.

**Ablation Study** First note that VES, which models only Gaussian data with linear dependencies, can be seen as an ablation where RNN and normalizing flow are not used. Next, to evaluate the usefulness of the normalizing flow in accurately modelling real world data, we analyze the performance of two particular instances of `NKF` : (i) '$f_t = $ id': the normalizing flow is set to be the identity function, as in Section 4.1, therefore also reducing to the `DeepState` model proposed in [9] (ii) '$f_t$ Local': the normalizing flow is applied *locally* for each time series, modelling non-Gaussianity and non-linearity, but not any dependencies between time series, as explained in Appendix C.2.2. This ablation is important in order to analyze the advantages of capturing the potential dependencies across time series in the data. Overall, we observe (bottom lines in Table 2) a significant increase in performance from the identity function to a local NF, along with another increase when applying the global NF (see `NKF` results), apart from the `wiki` dataset. While we do not have a satisfying explanation, we speculate that this is due to modelling errors, the optimization algorithm, or simply because the time series may not exhibit much dependencies between each other.

**Missing Data Experiment** We evaluate our model in the context of varying degrees of missing data during training and evaluation. From `elec`, we remove $p = 10 + 20k, k = 0 \ldots, 4$, percent of the training data at random. This dataset, while realistic, is highly regular with clear seasonality patterns. The models are then evaluated on rolling windows, where the input range observations are missing with the same probability. In Fig. 3 and Appendix C.3, we report results for our approach, along with `DeepAR`, `GP-Copula`, `KVAE`. We observe that not only does `NKF` outperform other approaches by a large margin, but its error also increases slower than in other methods when the percent-

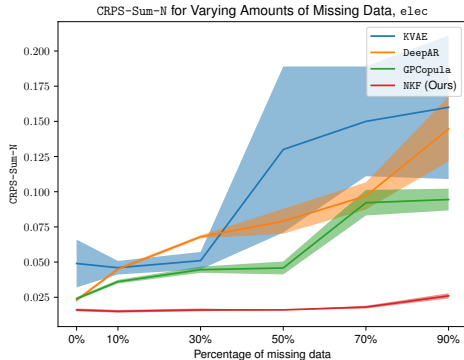

Figure 3: Forecasting with missing data.

age of missing data is increased. We believe that this is due to the proper handling of the uncertainties in our approach since our model does not directly take observations as input. Moreover, the strong results obtained for up to 90% of missing data demonstrate that our method encodes useful prior knowledge due to the structure induced in the `LGM`, rendering this method useful even in low data regimes (the same observation is made in [9] for this dataset).

## 5 Related Work

Neural networks for forecasting have seen growing attention in recent years [8, 27, 28, 29, 30, 31, 32]. We refer to [33] for an introductory overview. Most work concerns the univariate case using global models, assuming that time series are independent given the covariates and the model parameters. The family of global/local models, e.g., [34, 28], provide a more explicit way of incorporating global effects into univariate time series, without attempting to estimate the covariance structure of the data. An explicit probabilistic multivariate forecasting model is proposed in [1], which relies on Gaussian copulas to model non-Gaussian multivariate data.

Further related work combines probabilistic time series models with neural networks (e.g., point/renewal processes and neural networks [35] or exponential smoothing based expressions [27]). We extend the approach in [9] which uses an RNN to parametrize a state space model to the multivariate case alleviating Gaussianity and linear dependency assumptions in the observation model.

The idea to take advantage of the appealing properties of Kalman Filters (KF) [36] while relaxing its assumptions is not new. Prominent examples include the Extended Kalman Filter (EKF) [17], the Unscented Kalman Filter (UKF) [18] and Particle Filters (PF) [37] that relax the linearity and Gaussianity assumption by approximation or sampling techniques. The Gaussian process state space model (GPSSM) [38, 39] is a nonlinear dynamical system that extends LGMs by using GPs as the transition and/or observation mappings but typically assume the noise is additive Gaussian. If the noise is non-Gaussian, then these models again have to resort to approximation techniques similar to Particle Filters. Kernel Kalman Filters [40] address the linearity limitation of LGMs by defining the state space model in the reproducing Kernel Hilbert space (RKHS). In particular, the random latent state and the observation variable are mapped to RKHS and the state dynamics and the observation model are assumed to be linear in the kernel space. Note, however, that this approach still relies on the assumption that the noise is additive Gaussian.

Similarly, combining KF with neural networks is not new. Additionally to [9], [11] proposes to combine KF with Variational Auto-Encoders (KVAE) and [12] proposes variational approximations of the predictive distribution in nonlinear state space models. Finally, while most work on normalizing flows [13, 14, 41, 15] was presented in the i.i.d. setting, extensions to sequential data have recently been proposed [42, 43, 44]. Concurrent independent work by [45] also addresses the multivariate time series forecasting problem by combining normalizing flows with (deep) autoregressive models instead of state space models as in our work.

## 6 Discussion and Conclusion

In this paper we presented a simple, tractable and scalable approach to high-dimensional multivariate time series analysis, combining classical state space models with normalizing flows. Our approach can capture non-linear dependencies in the data and non-Gaussian noise, while still inheriting important analytic properties of the linear Gaussian state space model. This model is flexible, while still retaining interesting prior structural information, paramount to good generalization in low data regimes. Experimentally, our approach achieves the best results among a wide panel of competing methods on the tasks of forecasting and missing value handling. One caveat of our approach is that we no longer have identifiability w.r.t. the state space parameters: an interesting avenue of research is to work towards identifiability, e.g. by constraining the normalizing flow's expressivity.

## Broader Impact

The present article stems from the authors' work on time series forecasting and anomaly detection in industrial settings. The methods proposed here are not tied to specific time series applications, but will likely be beneficial in supply chain and monitoring settings where large panels of time series data are commonly produced, data generation processes are too complex to be modelled fully and full automation is an aspirational goal.

Beneficiaries of applications of this work are therefore primarily companies with data gathering infrastructure and historical data. Such companies will be able to make their processes more efficient given better time series analytics as proposed here. Societal consequences will be similar to other cases where resources can be put to a more efficient usage: less waste, lower energy consumption, less need for human intervention. While this sounds generally appealing, e.g., from an environmental perspective, there is a price for increased efficiency which we are observing in the present time: lack of robustness in the face of disaster.

The current COVID-19 crisis has revealed how overly lean supply chains (e.g., for medical supplies) can result in shortages. This phenomenon is not new and has been observed in the automotive industry for example in 2011 after an earthquake and tsunami stroke in Japan [46]. We speculate that such phenomena may occur in other application scenarios as well where the reduction of buffers is enabled through better predictive accuracies as presented here at the consequence of worse disaster recovery (e.g., more efficient usage of cloud compute resources). A large discussion in society is needed how we should balance efficiency and robustness in critical areas and the identification of these critical areas.

The central assumption in our methodology is that the past is a meaningful indication of the future. This assumption is, when disaster occurs, violated. Hence, systems relying on methods as ours need to handle such violations gracefully (see [47] for an early example). At present, human intervention and overrides must be enabled in systems incorporating our method. This central assumption also means that potential biases in the data will be reproduced unless otherwise intervened.

Finally, we remark that multivariate time series models may be attractive to model epidemics as the present and that it may be tempting to try out our method on the high-dimensional data currently observed. We strongly advise against drawing conclusions from such experiments. The spreading of diseases is a well-understood process and interventions such as lock-downs need to be properly modelled and accounted for. Much further work is needed to allow such fine-grained analysis with our method and a naive application of the present method will almost surely result in unwanted results and unnecessary confusion.

## Footnotes

[3]Placing the noise before the non-linearity $f_t$ in the observation model is important to obtain tractability for filtering and smoothing. However, this does not imply that data generated from a process where additive noise is added after the non-linear function cannot be modelled; in fact we use this particular model (nonlinear transformation with additive non-Gaussian noise) to generate data and test our method in the qualitative experiments in Section 4.1 (refer to appendix C.1 in the supplementary material for details).

[4]In Sec. 3, the complexity of the inverse is the same as the forward map, and the Jacobian term is linear in $N$.

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
