[Supplementary Material]

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

# Appendices

## A  Proofs

For completeness, we restate the `NKF` model:

$$
\begin{aligned}
\mathbf{l}_t &= F_t \mathbf{l}_{t-1} + \boldsymbol{\epsilon}_t, \quad \boldsymbol{\epsilon}_t \sim \mathcal{N}(0, \Sigma_t), \\
\mathbf{y}_t &= f_t(A_t^T \mathbf{l}_t + \boldsymbol{\varepsilon}_t), \quad \boldsymbol{\varepsilon}_t \sim \mathcal{N}(0, \Gamma_t).
\end{aligned}
\tag{9}
$$

### A.1  Filtering

**Proposition 1** (Filtering). The *filtered* distributions of the `NKF` model are *Gaussian* and are given by the filtered distributions of the corresponding `LGM` with pseudo-observations $z_t := f_t^{-1}(y_t)$, $t \geq 1$. That is, $p(l_t|y_{1:t}; \Theta, \Lambda) = p_{\texttt{LGM}}(l_t|z_{1:t}; \Theta)$ where $p_{\texttt{LGM}}$ refers to the distribution given by the `LGM`.

*Proof.* Note that for simplicity we omit conditioning on the parameters. Let us first recall the recursive Bayesian estimation, consisting of two distinct steps, predict and update of the latent state $l_t$ given by

$$
\begin{aligned}
\text{predict:} \quad & p(l_t|y_{1:t-1}) = \int p(l_t|l_{t-1})p(l_{t-1}|y_{1:t-1})\mathrm{d}l_{t-1}, \\
\text{update:} \quad & p(l_t|y_{1:t}) = \frac{p(y_t|l_t)p(l_t|y_{1:t-1})}{\int p(y_t|l_t)p(l_t|y_{1:t-1})\mathrm{d}l_t}.
\end{aligned}
\tag{10}
$$

The filtered distribution $p(l_t|y_{1:t})$ is obtained recursively applying both these steps until time $t$, starting from a prior on the state $p(l_1)$. However, in its current form one could expect that computing the latter would require approximating integrals, as equation 9 is non-linear and non-Gaussian, which is prohibitive in the high-dimensional setting. But we show by induction that filtered distributions are Gaussian and in fact coincide with the filtered distributions of the underlying linear Gaussian state space model.

Let $z_t := A_t l_t + \varepsilon_t$ so that $z_t = f_t^{-1}(y_t), \forall t = 1, 2, \ldots, T$. Since $y_t$ is observed and $f_t$ is an invertible, deterministic function, we can view $z_t$ as the pseudo-observation generated from the underlying linear Gaussian state space model (LGM),

$$
\begin{aligned}
\mathbf{l}_t &= F_t \mathbf{l}_{t-1} + \boldsymbol{\epsilon}_t, \quad \boldsymbol{\epsilon}_t \sim \mathcal{N}(0, \Sigma_t), \\
\mathbf{z}_t &= A_t^T \mathbf{l}_t + \boldsymbol{\varepsilon}_t, \quad \boldsymbol{\varepsilon}_t \sim \mathcal{N}(0, \Gamma_t),
\end{aligned}
\tag{11}
$$

By making use of the change of variables formula, the likelihood term in the update step of (10) can be written as:

$$
p(y_t|l_t) = p_{\mathbf{z}_t}(z_t|l_t) Df_t^{-1}(y_t),
\tag{12}
$$

where $Dg(x) := |\det[\mathrm{Jac}_x(g)]|$ is the absolute value of the determinant of the Jacobian of $g$ evaluated at $x$. By Eq. (11), $p_{\mathbf{z}_t}(z_t|l_t)$ is the density of the Gaussian variable $\mathbf{z}_t = A_t^T l_t + \boldsymbol{\varepsilon}_t$ conditioned on $l_t$, and from this we obtain $\forall t = 1, 2, \ldots T$,

$$
p(y_t|l_t) = \mathcal{N}(z_t|\, A_t^T l_t, \Gamma_t) Df_t^{-1}(y_t).
\tag{13}
$$

We proceed with inductive proof, first showing that the filtered distribution for the base case $t = 1$, $p(l_1|y_1)$, is Gaussian and the same as that of the underlying LGM. For this, we start with the first prediction step $p(l_1) = \mathcal{N}(\mu_1, \Sigma_1)$, which is assumed to be Gaussian. From Eq. (12), assuming $Df_1^{-1}(y_1)$ is non-zero, we get:

$$
p(l_1|y_1) = \frac{p(l_1)p(y_1|l_1)}{\int p(l_1)p(y_1|l_1)dl_1} = \frac{p(l_1)p_{\mathbf{z}_1}(z_1|l_1)Df_1^{-1}(y_1)}{\int p(l_1)p_{\mathbf{z}_1}(z_1|l_1)Df_1^{-1}(y_1)dl_1} = \frac{p(l_1)p_{\mathbf{z}_1}(z_1|l_1)}{\int p(l_1)p_{\mathbf{z}_1}(z_1|l_1)dl_1} = p_{\mathrm{LGM}}(l_1|z_1).
$$

This is exactly the first filtered distribution for LGM with observation $z_1$ and is in fact Gaussian since $p(z_1|l_1)$ is Gaussian by Eq. (13) and $p(l_1)$ is assumed to be Gaussian. Let $\mu_1^f, \Sigma_1^f$ denote the mean and covariance of this filtered distribution. For the inductive step assume that $p(l_{t-1}|y_{1:t-1})$ is Gaussian with mean $\mu_{t-1}^f$ and variance $\Sigma_{t-1}^f$, and is the same as $p_{\mathrm{LGM}}(l_{t-1}|z_{1:t-1})$. We will prove that $p(l_t|y_{1:t}) = p_{\mathrm{LGM}}(l_t|z_{1:t})$.

As both $p(l_{t-1}|y_{1:t-1})$ and $p(l_t|l_{t-1}) = \mathcal{N}(l_t|F_t l_{t-1}, \Sigma_t)$ are Gaussian, it follows that the distribution obtained from the prediction step must also be Gaussian, i.e., $p(l_t|y_{1:t-1}) = \mathcal{N}(l_t|\mu_t^p, \Sigma_t^p), \forall t = 2, \ldots, T$, with:

$$
\begin{aligned}
\mu_t^p &= F_t \mu_{t-1}^f, \\
\Sigma_t^p &= F_t \Sigma_{t-1}^f F_t^T + \Sigma_t.
\end{aligned}
\tag{14}
$$

In fact, this coincides with $p_{\mathrm{LGM}}(l_t|z_{1:t-1})$ given that $\mu_{t-1}^f, \Sigma_{t-1}^f$ are the same for both LGM and our model and both use the same transition for the latent state.

Similar to the base case, the filtered distribution for $t > 1$ is

$$
\begin{aligned}
p(l_t|y_{1:t}) &= \frac{p(y_t|l_t)p(l_t|y_{1:t-1})}{\int p(y_t|l_t)p(l_t|y_{1:t-1})dl_t} \\
&= \frac{Df_t^{-1}(y_t)p_{\mathbf{z}_t}(z_t|l_t)p(l_t|y_{1:t-1})}{\int Df_t^{-1}(y_t)p_{\mathbf{z}_t}(f_t^{-1}(y_t)|l_t)p(l_t|y_{1:t-1})dl_t} \\
&= \frac{p_{\mathbf{z}_t}(z_t|l_t)p_{\mathrm{LGM}}(l_t|z_{1:t-1})}{\int p_{\mathbf{z}_t}(z_t|l_t)p_{\mathrm{LGM}}(l_t|z_{1:t-1})dl_t} \\
&= p_{\mathrm{LGM}}(l_t|z_{1:t}),
\end{aligned}
\tag{15}
$$

which is the same as the filtered distribution of the corresponding LGM. In fact, one can deduce this filtered distribution in closed-form:

$$
\begin{aligned}
p(l_t|y_{1:t}) &= \frac{\mathcal{N}(z_t|\, A_t^T l_t, \Gamma_t)\mathcal{N}(l_t|\mu_t^p, \Sigma_t^p)}{\int \mathcal{N}(z_t|\, A_t^T l_t, \Gamma_t)\mathcal{N}(l_t|\mu_t^p, \Sigma_t^p)dl_t} \\
&= \mathcal{N}(l_t|\, \mu_t^f, \Sigma_t^f),
\end{aligned}
\tag{16}
$$

where $\mu_t^f = \mu_t^p + K_t[f^{-1}(y_t) - A_t^T \mu_t^p]$, $\Sigma_t^f = (I - K_t A_t^T)\Sigma_t^p$, and $K_t = \Sigma_t^p A_t(A_t^T \Sigma_t^p A_t + \Gamma_t)^{-1}$. This recursive formula for the filtered distribution is valid for $t > 1$ and the same for the base case $t = 1$ is obtained by noting that $\mu_1^p = \mu_1$ and $\Sigma_1^p = \Sigma_1$. $\qquad\square$

## A.2 Smoothing

**Proposition 2** (Smoothing). The *smoothed* distributions of the NKF model are *Gaussian* and are given by the smoothed distributions of the corresponding LGM with pseudo-observations $z_t := f_t^{-1}(y_t)$, $t = 1, 2 \ldots, T$. That is, $p(l_t|y_{1:T}; \Theta, \Lambda) = p_{\text{LGM}}(l_t|z_{1:T}; \Theta)$.

*Proof.* Note that for simplicity, we omit conditioning on the parameters. This can again be proved by induction starting with the base case $t = T$ and running backwards. For the base case the smoothed distribution $p(l_T|y_{1:T})$ coincides with the filtered distribution for the final time step $T$, which was already shown to be equal to that of the standard LGM. For the inductive step assume that in time step $t + 1$ it holds that $p(l_{t+1}|y_{1:T}) = p_{\text{LGM}}(l_{t+1}|z_{1:T})$. We will prove that the same is true in time step $t$, i.e., $p(l_t|y_{1:T}) = p_{\text{LGM}}(l_t|z_{1:T})$.

$$
\begin{aligned}
p(l_t|y_{1:T}) &= \int p(l_t, l_{t+1}|y_{1:T}) \mathrm{d}l_{t+1} \\
&= \int p(l_{t+1}|y_{1:T}) p(l_t|l_{t+1}, y_{1:T}) \mathrm{d}l_{t+1} \\
&= \int p(l_{t+1}|y_{1:T}) p(l_t|l_{t+1}, y_{1:t}) \mathrm{d}l_{t+1} \\
&= p(l_t|y_{1:t}) \int \frac{p(l_{t+1}|y_{1:T}) p(l_{t+1}|l_t)}{p(l_{t+1}|y_{1:t})} \mathrm{d}l_{t+1} \\
&= p_{\text{LGM}}(l_t|z_{1:t}) \int \frac{p_{\text{LGM}}(l_{t+1}|z_{1:T}) p(l_{t+1}|l_t)}{p_{\text{LGM}}(l_{t+1}|z_{1:t})} \mathrm{d}l_{t+1} \\
&= p_{\text{LGM}}(l_t|z_{1:T}),
\end{aligned}
\tag{17}
$$

where in the penultimate step we used the fact that the predictive distribution $p(l_{t+1}|y_{1:t})$ and the filtered distribution $p(l_t|y_{1:t})$ of our model are the same as those of the standard LGM (see Proposition 1 and (14)). The smoothed distribution of the standard LGM is Gaussian with mean $\mu_t^s$ and variance $\Sigma_t^s$ such that, starting from $\mu_T^s = \mu_T^f, \Sigma_T^s = \Sigma_T^f$:

$$
\mu_t^s = \mu_t^f + G_t[\mu_{t+1}^s - \mu_{t+1}^p], \tag{18a}
$$

$$
\Sigma_t^s = \Sigma_t^f + G_t[\Sigma_{t+1}^s - \Sigma_{t+1}^p], \tag{18b}
$$

$$
G_t = \Sigma_t^f A_t [\Sigma_{t+1}^p]^{-1}, \tag{18c}
$$

where $\mu_t^f, \Sigma_t^f$ correspond to mean and covariance computed during the update step of the NKF, and $\mu_t^p, \Sigma_t^p$ computed in the prediction step (refer to Eq. (14)).

$\square$

## A.3 Likelihood

**Proposition 3** (Likelihood). The likelihood of the parameters $(\Theta, \Lambda)$ of the NKF model given the observations $\{y_{1:T}\}$ can be computed as

$$
\ell(\Theta, \Lambda) = p(y_{1:T}; \Theta, \Lambda) = \prod_{t=1}^{T} p_{\text{LGM}}(z_t|z_{1:t-1}; \Theta) \left| \det \left[ \text{Jac}_{z_t}(f_t) \right] \right|^{-1}, \tag{4}
$$

where $z_t = f_t^{-1}(y_t)$ and $p_{\text{LGM}}(z_t|z_{1:t-1}; \Theta)$ denotes the predictive distribution of LGM.

*Proof.* We can compute the likelihood by decomposing it into telescoping conditional distributions and using the substitution $z_t = f^{-1}(y_t)$,

$$
\begin{aligned}
p(y_{1:T}; \Theta, \Lambda) &= \prod_{t=1}^{T} p(y_t | y_{1:t-1}; \Theta, \Lambda) \\
&= \prod_{t=1}^{T} \int \left| \det \left[ \mathrm{Jac}_{y_t}(f_t^{-1}) \right] \right| p_{\mathbf{z}_t}(z_t | l_t) p(l_t | y_{1:t-1}; \Theta) \mathrm{d}l_t \\
&= \prod_{t=1}^{T} \left| \det \left[ \mathrm{Jac}_{y_t}(f_t^{-1}) \right] \right| \int p_{\mathbf{z}_t}(z_t | l_t) p_{\mathtt{LGM}}(l_t | z_{1:t-1}; \Theta) \mathrm{d}l_t \qquad (19) \\
&= \prod_{t=1}^{T} \left| \det \left[ \mathrm{Jac}_{y_t}(f_t^{-1}) \right] \right| p_{\mathtt{LGM}}(z_t | z_{1:t-1}; \Theta) \\
&= \prod_{t=1}^{T} \left| \det \left[ \mathrm{Jac}_{y_t}(f_t^{-1}) \right] \right| \mathcal{N}(z_t; \nu_t^p, \Gamma_t^p),
\end{aligned}
$$

where in the third step we used the fact that predictive distributions of our model (9) and the standard LGM are the same. This is true because the filtered distributions of our model and the LGM are the same since both use the same transition for the latent state, as shown in Proposition 1 (see (14)). In the final step we used the fact that the predictive distribution $p(z_t | z_{1:t-1})$ of the standard LGM is Gaussian with mean $\nu_t^p$ and covariance $\Gamma_t^p$ given by the analytical expressions [19]:

$$
\begin{aligned}
\nu_t^p &= A_t^T F_t \mu_{t-1}^f, & \Gamma_t^p &= A_t^T \left( F_t \Sigma_{t-1}^f F_t^T + \Sigma_t \right) A_t + \Gamma_t, & t &> 1, \\
\nu_1^p &= A_t^T \mu_1, & \Gamma_1^p &= A_t^T \Sigma_1 A_t + \Gamma_1, & t &= 1.
\end{aligned}
$$

$\square$

### A.4 Forecasting Distribution

The joint forecasting distribution of the future time steps $T + 1 : T + \tau$ can be obtained in terms of the predictive distribution of the corresponding LGM with $z_t = f^{-1}(y_t)$,

$$
p(y_{T+1:T+\tau} | y_{1:T}, x_{1:T+\tau}; \Theta, \Lambda) = \prod_{t=T+1}^{T+\tau} p_{\mathtt{LGM}}(z_t | z_{1:t-1}; \Theta) \left| \det \left[ \mathrm{Jac}_{z_t}(f_t) \right] \right|^{-1}. \qquad (20)
$$

The exact analytical expressions for the forecasting distribution is given by:

$$
p(y_{T+1}, \ldots, y_{T+\tau} | y_1, \ldots, y_T, \Theta) = \prod_{t=T+1}^{T+\tau} \mathcal{N}(f^{-1}(y_t); \nu_t^p, \Gamma_t^p) \left| \det \left[ \mathrm{Jac}_{y_t}(f_t^{-1}) \right] \right|, \qquad (21)
$$

$$
\begin{aligned}
\nu_t^p &= A_t^T \mu_t^p, & \mu_t^p &= F_t \mu_{t-1}^f, \qquad (22) \\
\Gamma_t^p &= A_t^T \Sigma_t^p A_t + \Gamma_t, & \Sigma_t^p &= F_t \Sigma_{t-1}^f F_t^T + \Sigma_t, \qquad (23)
\end{aligned}
$$

where $(\mu_t^p, \Sigma_t^p)$ and $(\nu_t^p, \Gamma_t^p)$ are the parameters of the predictive distributions for the latent state and observations given in Propositions 1 and 3, respectively.

### A.5 Handling Missing Data

Given a subset of observed targets $y_{\mathbf{t}^{\mathrm{obs}}}$, what can we say about the missing observations? Said otherwise, what is the probability of $p(y | y_{\mathbf{t}^{\mathrm{obs}}})$? This data imputation problem is of central importance

in numerous applications. This problem can be solved by first solving the smoothing problem in order to obtain the posterior $p(l_t|y_{\mathbf{t}^{\mathrm{obs}}})$:

$$
\begin{aligned}
p(y|y_{\mathbf{t}^{\mathrm{obs}}}) &= \int p(y, l|y_{\mathbf{t}^{\mathrm{obs}}})\mathrm{d}l \\
&= \int p(y|l)p(l|y_{\mathbf{t}^{\mathrm{obs}}})\mathrm{d}l \\
&= \prod_t p_{\mathtt{LGM}}(z_t|z_{\mathbf{t}^{\mathrm{obs}}}) \left|\det\left[\mathrm{Jac}_{z_t}(f_t)\right]\right|^{-1} \\
&= \prod_t \mathcal{N}(z_t|A_t^T \mu_t^{u,s}, A_t^T \Sigma_t^{u,s} A_t + \Gamma_t) \left|\det\left[\mathrm{Jac}_{z_t}(f_t)\right]\right|^{-1},
\end{aligned}
\tag{24}
$$

where $(\mu_t^{u,s}, \Sigma_t^{u,s})$ corresponds to the parameters of the smoothed distribution in the presence of missing data. Once again, this distribution admits an analytical expression and can be readily sampled from.

# B  Model: Additional Details

## B.1  Encoding of the LGM Parameters

In order to constrain the real-valued outputs $h_t$ at time $t$ of the RNN $\Psi$ to the parameter domains of the LGM, we apply a sequence of transformations. For the $j$-th state space model parameter, $\Theta_t^j$, we initially compute the affine transformation $\tilde{\Theta}_t^j = w_j^\top h_t + b_j$, where the weights and biases are different for each parameter and are all included in $\Phi$ and learned. We then transform $\tilde{\Theta}_t^j$ to the domain of the parameter by applying:

- for real-valued parameters: no transformation i.e., $\Theta_t^j = \tilde{\Theta}_t^j$,
- for positive parameters: the softplus function $\Theta_t^j = \log(1 + \exp(\tilde{\Theta}_t^j))$,
- for bounded parameters in $[a, b]$: a scaled and shifted sigmoid $\Theta_t^j = (b - a)\frac{1}{1+\exp(-\tilde{\Theta}_t^j)} + a$.

In practice, it is often advisable to impose stricter bounds than theoretically required, e.g., enforcing an upper bound on the observation noise variance or a lower bound on the innovation strengths can stabilize the training procedure in the presence of outliers.

## B.2  Local-Global Instantiation

As in [9], the evolution of the latent state $\mathbf{l}_t^{(i)}$ for each time series $i$ is captured using a *composition* of level-trend and seasonality model, described below.

**Local Level-Trend Model**  In the level-trend model, the latent state has two dimensions and is characterized by:

$$
\begin{aligned}
F_t^{(i)} &= \begin{bmatrix} 1 & 1 \\ 0 & 1 \end{bmatrix}, \quad A_t^{(i)} = \begin{bmatrix} 1 \\ 1 \end{bmatrix}, \\
\Sigma_t^{(i)} &= \begin{bmatrix} \alpha_t^2 & 0 \\ 0 & \beta_t^2 \end{bmatrix}, \quad \Gamma_t^{(i)} \in \mathbb{R}_{++}.
\end{aligned}
\tag{25}
$$

**Local Seasonality Model**  In the case of seasonality-based models, each seasonality pattern can be described by a set of seasonal factors (or seasons) associated with it. For example, in the day-of-week pattern there are seven factors, one for each day of the week. We can represent each factor as a component of the latent state $\mathbf{l}_t \in \mathbb{R}^7$. Then, for the day-of-week seasonality model, we have

$$
\begin{aligned}
F_t^{(i)} &= I, \quad A_t^{(i)} = \mathbb{1}_{\{\mathrm{day(t)=j}\}_{j=1}^7}, \\
\Sigma_t^{(i)} &= \sigma_t^2 \mathrm{diag}(A_t^{(i)}), \quad \Gamma_t^{(i)} \in \mathbb{R}_{++}.
\end{aligned}
\tag{26}
$$

**Composite Model**  We concatenate the state of the level-trend model and the seasonality model in order to take into account both types of dynamics.

Note that in order to take into account the correlations across time series, the *pseudo-observations* generated by the composite state space model are given to the normalizing flow $f_t$, which will implicitly capture these correlations.

## C  Experimental Details

We use the same hyperparameters for the model architecture across all datasets. For the RNN, we use an LSTM with the same architecture and hyperparameters as those proposed in `DeepState` [9], based on the open-source implementation from [48]. To avoid numerical issues arising during the `NKF` update step, we find it useful to lower bound the observation noise $\Gamma_t^{(i)}$ to $0.1$ for `elec`, `solar`, `wiki`, and $0.01$ for the rest. The numerical issues are perhaps due to the overfitting of LGM parameters to the training data; the open-source implementation [48] of `DeepState` [9] also recommends such safe guards on the noise terms. Note that in the experiments $f_t$ is the same across all time steps; however time-dependant noise may still be captured as the parameters of the LGM are time-dependent. An interesting research avenue for future work may be to consider a time-varying normalizing flow by conditioning it on temporal features, thus bringing us to consider conditional NFs [10] .

We evaluated various NF proposed in the literature: *RealNVP* [13], *Glow* [14] and *iResnet* [41]. We have finally opted for RealNVP which is straightforward to implement and does not suffer from drawbacks of the other methods. In particular, the number of parameters in Glow scales quadratically in the number of dimensions due to the fully connected layers in the permutation step: for the wiki dataset, as the dimension of the observations is $2 \times 10^3$, this would imply that just one permutation layer would have $4 \times 10^6$ parameters. In iResnet, the estimator of the Jacobian term yielded a high variance in the high dimensional setting and requires a number of forward passes in order to compute the inverse. For all the experiments, we have set the number of blocks of the RealNVP to $9$. Each affine-coupling is parameterized by a 2-layer neural network with the numbers of hidden dimensions set to $16$, without batch-normalization.

During training we tune the number of epochs for each dataset on a separate validation set of equal size to the test set. We use the Adam optimizer with $2 \times 10^{-4}$ learning rate and $1 \times 10^{-6}$ weight-decay.

As in [1], the evaluation is done in a rolling fashion: for hourly datasets accuracy is measured on 7 rolling time windows where each roll corresponds to 24 hours, thus covering 7 days of the test set. For all the other datasets we use 5 windows. More details on the forecast horizon $\tau$, domain, frequency, dimension $N$ and length of training timesteps $T$ are given in the Appendix C.2.1.

### C.1  Qualitative Experiments

#### C.1.1  Datasets

Both datasets are composed of 2 daily univariate time series with weekly seasonality, correlated in different ways. The time series observations $y_{1:T}$, with $T = 120$, are generated according to the following state space model:

$$\mathbf{l}_t = \mathbf{l}_{t-1} + \boldsymbol{\epsilon}_t, \quad \boldsymbol{\epsilon}_t \sim \mathcal{N}(0, 10^{-3} \times I),$$
$$\mathbf{y}_t = m(A_t^T \mathbf{l}_t) + \boldsymbol{\eta}_t, \quad \boldsymbol{\eta}_t \sim \mathcal{D}(t), \tag{27}$$

where $\mathbf{l}_t = [\mathbf{l}_t^{(1)}; \mathbf{l}_t^{(2)}] \in \mathbb{R}^{14}$ and each component of $\mathbf{l}_t^{(i)} \in \mathbb{R}^7$ is associated to a day of the week, $A_t \in \mathbb{R}^{14 \times 2}$ is block diagonal where $A_t^{(i)} = \mathbb{1}_{\{\text{day(t)=j}\}_{j=1}^7}$ *selects* the corresponding state component based on $t$, $m : \mathbb{R}^2 \to \mathbb{R}^2$ is a *mixing* function, and $\mathcal{D}(t)$ is a highly non-Gaussian, time-dependent distribution. The initial state $l_1^{(i)}$ is sampled from a Gaussian distribution $\mathcal{N}(\mu_0, 10^{-3} \times I)$, with 3 different mean levels according to the day of the week[5]: $\mu_1 = [-10, -10, 0, 0, 0, 10, 10]^T$.

For the first experiment (corresponding to Figure 2a) we consider the simple case where no mixing occurs, i.e., $m$ is the identity function. The observational noise $\boldsymbol{\eta}_t$ is the same for every $t$ and highly

non-Gaussian. To this respect, $\eta_t^{(1)}$ follows an 1D Uniform distribution and $\eta_t^{(2)}$ is the image of $\eta_t^{(1)}$ when mapped through a cosine function.

For the second experiment (corresponding to Figure 2b) we consider a more complex setting where mixing occurs and the observational noise $\eta_t$ is time-dependent. Artificial dependencies across time series are induced, setting the *mixing* function $m$ to $m([x,y]^T) = [x\,y, x+y]^T$. The observational noise $\eta_t$ follows a day-of-week pattern, where three non-Gaussian distributions are selected based on the day of the week, similarly as for $\mu_1$: for the first two days $\eta_t$ is the same in the first experiment, for the three next days $\eta_t$ follows a mixture of two Gaussians, and for the weekend $\eta_t$ is a simple line generated from an 1D Uniform distribution.

### C.1.2 Forecast Samples

Figure 4: Forecast results with NF (`NKF` model) for both time-series of dataset `S1` (left) and `S2` (right). For each forecast: In blue, the values of the input (left of red line) and target (right of red line) time series. In dark green, the forecasts mean, and quantiles $[0.1, 0.9]$ are filled light green.

## C.2 Quantitative Experiments

### C.2.1 Datasets

We evaluate on the public datasets (with the same training and test splits) used in [1]: `exchange`: daily exchange rate between 8 currencies as used in [49]; `solar`: hourly photo-voltaic production of 137 stations in Alabama State used in [49]; `elec`: hourly time series of the electricity consumption of 370 customers [50]; `traffic`: hourly occupancy rate, between 0 and 1, of 963 San Francisco car lanes [50]; `wiki`: daily page views of 2000 Wikipedia pages used in [29]. Similar to [1], the forecasts of different methods are evaluated by splitting each dataset in the following fashion: all data prior to a fixed *forecast start date* compose the training set and the remainder is used as the test set. We measure the accuracy of the forecasts of various methods on all the time points of the test set.

In Table 3 we summarize the details of the datasets used to evaluate the models.

### C.2.2 Ablation Study: $f_t$ `Local`

Here we give additional details for the local variant of our model `NKF-Local`. This variant uses the same form for the state-space model (Eq. (8a) and (8b)), with an alternative local normalizing flow $u_t : \mathbb{R} \to \mathbb{R}$, applied to each time-series independently:

$$f_t(z_t) = (u_t(z_{1,t}), \dots, u_t(z_{N,t})). \tag{28}$$

| dataset | $\tau$ (num steps predicted) | domain | frequency | dimension $N$ | time steps $T$ |
|---|---|---|---|---|---|
| exchange | 30 | $\mathbb{R}^+$ | daily | 8 | 6071 |
| solar | 24 | $\mathbb{R}^+$ | hourly | 137 | 7009 |
| elec | 24 | $\mathbb{R}^+$ | hourly | 370 | 5790 |
| traffic | 24 | $\mathbb{R}^+$ | hourly | 963 | 10413 |
| wiki | 30 | $\mathbb{N}$ | daily | 2000 | 792 |

Table 3: Summary of the datasets used to test the models. Number of steps forecasted, data domain $\mathcal{D}$, frequency of observations, dimension of series $N$, and number of time steps $T$.

In this case, the conditional density in Eq. (2) of variables formula can easily be expressed in terms of normalizing flow $u_t$, and reduces to:

$$
\begin{aligned}
p(y_t|l_t; \Theta, \Lambda) &= p(y_{1,t}, \ldots, y_{N,t}|l_t; \Theta, \Lambda) \\
&= p_{z_t}(u_t^{-1}(y_{1,t}), \ldots, u_t^{-1}(y_{N,t}); \Theta) \prod_{i=1}^{N} \left| \det \left[ \mathrm{Jac}_{y_{t,i}}(u_t^{-1}) \right] \right|.
\end{aligned}
\tag{29}
$$

Written in this form, we can see that the computation of the Jacobian term scales linearly in the dimension, as $y_{i,t} \in \mathbb{R}$ and can be calculated in parallel.

For the univariate $u_t$, we use non-time dependent iResnet [41], with 6 invertible blocks, LipSwish activation, spectral normalizing coefficient of 0.9, and 10 fixed-point iterations for the computation of the inverse, and 1 iteration for the power iteration method. We set the learning rate associated to the RNN to 0.001, and 0.00001 for the normalizing flow.

## C.3 Handling Missing Data

Here we report the exact numbers for the missing data experiment, Table 4.

|  | elec10% | elec30% | elec50% | elec70% | elec90% |
|---|---|---|---|---|---|
| KVAE | $0.046 \pm 0.0048$ | $0.051 \pm 0.0062$ | $0.13 \pm 0.059$ | $0.15 \pm 0.039$ | $0.16 \pm 0.051$ |
| DeepAR | $0.045 \pm 0.001$ | $0.068 \pm 0.001$ | $0.079 \pm 0.009$ | $0.097 \pm 0.010$ | $0.145 \pm 0.023$ |
| GP-Copula | $0.036 \pm 0.0013$ | $0.045 \pm 0.002$ | $0.046 \pm 0.0046$ | $0.092 \pm 0.0091$ | $0.094 \pm 0.0077$ |
| NKF | $\mathbf{0.015 \pm 0.001}$ | $\mathbf{0.016 \pm 0.001}$ | $\mathbf{0.016 \pm 0.000}$ | $\mathbf{0.018 \pm 0.001}$ | $\mathbf{0.026 \pm 0.0019}$ |

Table 4: CRPS-Sum-N (lower is better), averaged over 3 runs.

## C.4 A Note on Evaluation Multivariate Metrics for Probabilistic Forecasting

CRPS relies on the pinball loss that measures the fit at each quantile between the quantile function $F^{-1}$ and an observation:

$$
\Lambda_\alpha(q, y) = (\alpha - \mathbb{1}_{\{y < q\}})(y - q),
\tag{30}
$$

where $\alpha \in [0, 1]$ is the quantile level and $q$ the respective quantile of the probability distribution.

The integrated pinball loss over all quantile levels $\alpha \in [0, 1]$ is defined as the CRPS:

$$
\mathrm{CRPS}(F^{-1}, y) = \int_0^1 2\Lambda_\alpha(F^{-1}(\alpha), y)\, d\alpha.
\tag{31}
$$

We estimate $F^{-1}$ by drawing 100 samples and sorting them.

Although CRPS is a widely accepted metric for assessing the quality of probabilistic forecasts in the univariate case, it is not applicable in the multivariate setting as it does not capture correlations across time-series. Instead, [1] introduced CRPS-Sum an extension of CRPS to the multivariate case:

$$
\mathrm{E}_t[\mathrm{CRPS}(F^{-1}, \sum_i y_{i,t})],
$$

where $F^{-1}$ is estimated by summing the samples across dimensions and then computing the quantiles by sorting. However, [1] does not give a theoretical justification of CRPS-Sum.

Below, we prove that CRPS-Sum is a proper scoring rule.

We restate fundamental definitions and results from [24] on proper scoring rules first before showing that CRPS-Sum-N is a proper scoring rule.

Let $\Omega$ be the sample space, let $\mathcal{A}$ be a $\sigma$-algebra on $\Omega$, and let $\mathcal{P}$ be a convex class of probability measures on $(\Omega, \mathcal{A})$.

**Definition 1.** A *scoring rule* is a function $S : \mathcal{P} \times \Omega \to [-\infty, \infty]$, such that for every $P \in \mathcal{P}$ we have that $E(S(P, \cdot))$ exists (and is possibly non-finite). For such an $S$, and for every $P, Q \in \mathcal{P}$, define

$$S(P, Q) := \int S(P, \cdot) dQ.$$

Properness is defined thusly:

**Definition 2.** A scoring rule is proper with respect to $\mathcal{P}' \subset \mathcal{P}$ if $\forall P, Q \in \mathcal{P}'$ we have that $S(P, P) \geq S(Q, P)$. It is called strictly proper with respect to $\mathcal{P}'$ if equality holds iff $P = Q$ a.s.

**Theorem 3.** *Let $\Omega = \mathbb{R}$, and $\mathcal{A}$ be the Borel $\sigma$-algebra. Then the function*

$$\mathrm{CRPS}(P, x) := -\int_{\mathbb{R}} (P(\{t | t \leq y\}) - \mathbb{1}_{x \leq y})^2 dy$$

*is a proper scoring rule with respect to the set $\mathcal{P}$ of probability measures on $\mathcal{A}$. Furthermore, if we restrict to the subclass $\mathcal{P}'$ of probability measures with finite first moment, then it can be shown that*

$$\mathrm{CRPS}(P, x) = \frac{1}{2} E_P(|X - X'|) - E_P(|X - x|),$$

*where $X$ and $X'$ are understood as independent random variables that have the distribution $P$, and* CRPS *is strictly proper with respect to $\mathcal{P}'$.*

**Note 4.**

1. The term "$E_P(|X - X'|)$ where $X$ and $X'$ are understood as independent random variables that have the distribution $P$" means, by definition: $\int_{\Omega \times \Omega} |p_1 - p_2| d(P \times P)$, where $p_i$ is the projection to the $i^{th}$ coordinate.

2. It is crucial in the definition of properness that we require that $\forall P, Q \in \mathcal{P}'$ we have that $S(P, P) \geq S(Q, P)$ rather than $S(P, P) \geq S(P, Q)$. Indeed, it is easy to see that if $P$ follows a standard normal distribution and $Q$ is the constant distribution $0$, then

$$\mathrm{CRPS}(P, Q) - \mathrm{CRPS}(P, P) =$$
$$E_P(|X - X'|) - E_P(|X|) =$$
$$\frac{2}{\sqrt{\pi}} - \sqrt{\frac{2}{\pi}} > 0.$$

3. The proof of strict properness in Theorem 3 with respect to $\mathcal{P}'$ boils down to the statement that for any independent $X$ and $X'$ following a probability distribution $P \in \mathcal{P}'$, and $Y$ and $Y'$ independent of each other and of $X$ and $X'$, following a probability distribution $Q \in \mathcal{P}'$, we have that:

$$2E(|X - Y|) - E(|X - X'|) - E(|Y - Y'|) =$$
$$\int (P(\{t | t \leq y\}) - Q(\{t | t \leq y\}))^2 dy,$$

and is therefore non-negative, and zero iff $P = Q$ a.s. An elementary proof can be found in pages 5 and 6 of [51].

In the case that $\Omega = \mathbb{R}^d$ and $\mathcal{A}$ is its associated Borel $\sigma$-algebra, we introduce the following new scoring rule.

**Definition 5.** For any choice of a measurable function $L : \mathbb{R}^d \to \mathbb{R}$ w.r.t to the Borel $\sigma$-algebras, define:

$$\text{CRPS}(L, P, x) := \text{CRPS}(L_* P, L(x)),$$

where $L_* P$ is the pushforward measure of $P$ by $L$.

It is trivial to verify that for every such choice of $L$, the function $\text{CRPS}(L, \cdot, \cdot)$ is a scoring rule. Also note that the scoring rule $CRPS - Sum - N$ from Appendix G.1 of [1] is none other than $\text{CRPS}(L, \cdot, \cdot)$ for $L$ defined by

$$L(x_1, ..., x_d) = x_1 + ... + x_d.$$

In what follows we will not restrict to this choice of $L$.

**Lemma 6.** *The following equality holds for all probability measures $P$ and $Q$:*

$$\text{CRPS}(L, P, Q) = \text{CRPS}(L_* P, L_* Q).$$

*Proof.* This boils down to the change-of-variables formula in measure theory:

$$\text{CRPS}(L, P, Q) = \int_{\mathbb{R}^d} \text{CRPS}(L_* P, L(x)) dQ =$$

$$\int_{\mathbb{R}} \text{CRPS}(L_* P, x) dL_* Q = \text{CRPS}(L_* P, L_* Q)$$

$\square$

Therefore, we get the easy corollary:

**Corollary 7.** $\text{CRPS}(L, \cdot, \cdot)$ *is proper with respect to the Borel measurable sets.*

*Proof.* For any two probabilty distributions on the Borel measurable sets on $\mathbb{R}^d$:

$$\text{CRPS}(L, Q, P) = \text{CRPS}(L_* Q, L_* P) \leq$$

$$\text{CRPS}(L_* P, L_* P) = \text{CRPS}(L, P, P).$$

$\square$

If $d > 1$, then for any reasonable large convex set of probability measures $\mathcal{P}'$ (e.g., the class of probabilty measures with finite first moments), and for any $P \in \mathcal{P}'$, it is always possible to find multiple $Q$'s in $\mathcal{P}'$ such that $L_* Q = L_* P$. Therefore $d > 1$ implies that $\text{CRPS}(L, \cdot, \cdot)$ is proper but not strictly proper.