[Reviews · NeurIPS 2020]

Review 1

Summary and Contributions: Linear, Gaussian state space models (like the Kalman filter) have huge things going for them. Trouble is they are linear and Gaussian! Over the years there have been many attempts to update the humble KF to avoid these limitations. This paper uses neural nets to do normalizing flows. The model created has all the benefits of the KF, like speed, scalability, dealing properly with missing and uncertain data, iterative inference over future points, uncertainty estimation baked in etc - but, crucially, avoid the big assumptions (and limitations) of things like the KF. the authors develop the method and showcase its performance on a variety of problems.

Strengths: Like so many good ideas, when I read this I could see how nice this was. It is a level above many of the approaches put forward over recent years to solve this kind of problem. The method seems profound, scalable and has novelty and neatness. It will be an interesting read for the NeurIPS community. * I checked all theory and found (to best of my ability) it entirely correct - offering a solid sounding to the method * the approach is well discussed and the audit trail of reasoning is particularly well made * the empirical testing is extensive and highlights well the innovation * the paper is very well-written, an enjoyable read and provides a great review of most of the current (and historical) methods as well as an excellent, clear description of the model proposed

Weaknesses: Not many weaknesses, though if I am being picky a few stand out: * there is no real discussion about the (to my mind clear) relation to - kernelized Kalman filters (which offer some similar benefits), and non-Gaussian AR methods - Gaussian process state space models (Simo Särkkä and others) - which although in principle retain some of the problems of 'linear combination' do so in a robust, and principled approach (similar to kernels). I also include in this some techniques, which could be regarded as a 'flow', such as GP latent variable model (Lawrence et al). * the tables of results are nicely presented, yet no significance testing is performed. For example we see values like 0.005 \pm 0 and 0.006 \pm 0.001. Are these truly different given the sample size?

Correctness: Yes - I checked all the development of the theory and, to the best of my ability, found everything in order. The empirical tests are quite extensive, though my comments about state space GP inclusion and significance tests still hold)

Clarity: The paper is particularly well-written, easy and enjoyable to read. The method is presented clearly along with review and discussion of other approaches.

Relation to Prior Work: Yes - in that this way of doing things has not been published before (best of my searching & knowledge). The paper does explain clearly how the method relates to several other approaches, some similar and others not so. Comments about kernel-KF and SS-GP as above.

Reproducibility: Yes

Additional Feedback: Nice paper - really enjoyed reading this. A few minor comments beyond those mentioned in other sections. * Prop 3. I see what you mean referring to the EKF as a nonlinear extension of the LGM. But, unlike the UKF, it's kind of just a linear model but with an augmented state space (with gradients) * 'Forecasting and Missing Values' - the hyperparams ('observation' and 'state' noise variances) are sampled. Seems like a variational approximation would be very tight here - have you considered this to avoid sampling? - the missing samples approach seems the same as that proposed for the KF by Shumway and Stoffer. * maybe just me, but Fig. 2 took me a long time to figure out. Looks so much like a 2-d problem space rather than 1d in time. * Sec 4.2. Several cases of missing spaces between acronyms, cites and words. * References: protect your Capitals. {G}aussian, {GARCH} etc -- Dear Authors. Thank you for addressing all concerns raised. Whilst some issues still remain (questions regarding noise terms most notably) this is still a very good paper imho and I enjoyed the read.


Review 2

Summary and Contributions: The paper proposes a probabilistic model for multivariate time series, permitting nonlinear dependence between dimensions and across time. This is achieved via use of a normalizing flow (NF) for the emission of a SSM with time-dependent linear dynamics. The authors show that inference and learning in the model is analytically tractable, and demonstrate strong performance for forecasting in five public datasets.

Strengths: * The paper presents an important and useful observation that SSMs with a nonlinear output can admit analytic inference via use of NF-style emissions. * As a consequence, the full gamut of probabilistic machinery of a LGSSM (analytic updates, inference, smoothing, forecasting, missing value imputation) can be applied to a specific form of nonlinear model in an efficient manner. * The authors further propose that the parameters of the LGSSM can be time dependent, using an RNN for estimation as per e.g. the DeepAR and DSSM models. In soundbite form, the paper combines the benefits of the KVAE and DSSM models. * The experiments and discussion provide evidence that the NKF mitigates many of the flaws (linear assumptions, inefficient inference, handling missing data) of previous work, allowing results that are consistently strong (albeit perhaps not much better) for a variety of datasets.

Weaknesses: * The NF assumption was not discussed as compared to a standard SSM which uses additive measurement noise. Placing the emission noise *before* the nonlinearity is a crucial move; otherwise filtering is not tractable. I would have appreciated further discussion of the impact of this. It's possible that this technique can be applied as a drop-in replacement in many models to avoid awkward approximations such as EKF, UKF and PF; however this conclusion is not immediate from the work presented in this paper. * As a simple example, consider a univariate example where $f$ is a sigmoid, and the true $z = -5$, hence $\E[y] ≈ 0$. If $y$ is observed with additive noise of +0.2, the inferred $z = f^{-1}(0.2) ≈ -1.4$, which may cause substantial problems for inference and learning. * The qualitative experiments seemed particularly artificial; I did not learn much here beyond the fact that the implementation broadly seems to work. If these are indicative of a real-world problem, it would be helpful to make this clearer. * NKF does not show markedly better performance than the GP-Copula model in the main experiments.

Correctness: I did not encounter any important claim that I believe is incorrect. Three concerns relating to correctness are provided below and a variety of smaller concerns are provided in section 9. [L49]: "A thorough evaluation of applicability of normalizing flows". I think this should be weakened or at least qualified: (i) only the _performance_ is shown on some common tasks; the appropriateness (or applicability) is not carefully validated; (ii) a thorough evaluation of NF in general in a time series context should also investigate NF in the context of the transition model. [L269]: "inference and likelihood computation are not tractable in KVAE and it relies on particle filters for their approximation". I agree that KVAE relies on Monte Carlo samples for $z$, but uses KF steps, not particle filtering for the latent dynamics. Indeed, that is one of the main contributions of KVAE. [Table 2]: The standard deviations are _very_ small, which do not appear to be sensible reflections of the variability of model performance. This is clear in comparison to the results of [1]: the results of GP-Copula for some datasets differ between your paper and [1] by orders of magnitude more than the computed s.d.s. This makes assessment of significance difficult.

Clarity: The majority of the paper was easy and enjoyable to read. However I found Section 3 more difficult to follow. In particular I found para 2 (lines 193-204) confusing. E.g. "each time series is associated with latent factors that evolve independently w.r.t. the factors of the other time series" suggests a 1:1 correspondence between a time series and a latent factor. But I understand eq. (8) simply as a factorial state assumption, and each time series is a mixture of *all* $N$ latent factors via the NF in (8c)?

Relation to Prior Work: Yes. I am not aware of any further (strongly) related work. The key difference of tractable inference in nonlinear emission models without approximation is clearly stated.

Reproducibility: Yes

Additional Feedback: I very much enjoyed the paper, and I congratulate the authors on their work. The experimental results by themselves do not necessarily provide a compelling reason to use NKF over previous models. But I think the idea is an important and straight-forward one, addressing a variety of weaknesses of previous models. On the other hand, the omission of any discussion of the non-additive noise seemed problematic to me. Minor comments: [L262]: [5] is a textbook, please be more specific in the ref. Also "a direct generalization of linear state space model to multivariate time series..." is confusing. Linear SSMs are very commonly used to model multivariate time series. What is the meaning of "generalization"? [L267]: spaces missing after \texttt. [L283]: "We also did not choose log-likelihood since not all methods yield analytical forecast distributions and is not meaningful for some methods". To my mind, it would be helpful to also include a more familiar comparison than CRPS e.g. MSE, as in [1]. While MSE does not capture the calibration of the model likelihood, it's an important alternative view of performance. [L288]: "Classical methods... [may] fail with numerical problems". In appendix C, you mention that NKF also suffers from numerical problems, which you fixed by lower bounding the observation noise. In the light of this, is this a fair statement? [L304]: "[we observe]...another increase when applying the global NF". I disagree. It seems to me from Table 2 only the `solar` task showed meaningful improvement from the `Local` ablation model. `exchange` and `traffic` are not statistically different (even assuming the displayed CI is meaningful), and `elec` is borderline. I have no problem with this. The `Local` model is also a novel method so far as I understand. [L307]: EOL missing due to overlap with figure. Supplementary material: [L645]: "To avoid numerical issues... we find it useful to lower bound the observation noise". Why did certain datasets cause more substantial numerical problems? A possible reason may be that the RNN predicting the SSM parameters was overfitting to the training set causing a collapse of the covariance -- was this an issue? [L676]: Should this be $\mu_0$? [Figure 4]: Presumably these are two different samples of time series generated from S1/S2. It might be helpful to state this in the text. More importantly, it would be useful to provide a comparison of this time series view with the LGM version as in Fig 2(a). While Fig 2(a) shows that the distribution of the noise variable is captured well by the NKF, it's not necessarily obvious that this translates to better predictive accuracy, especially since quantitative results are omitted. ------------- I thank the authors for their response. I maintain the positive score from my initial review. I am still of the opinion that the non-additive noise term demands some discussion; the examples given in the response (Appx. C1) either have no nonlinearity or a simple multilinearity in the generative model. If, on the other hand, the generative process performs any strong 'squashing' of the probability density, then one may expect poorer performance of the underlying LGM. Hence I have insufficient evidence that the NKF can be used as a drop-in replacement for existing methods in all cases. But the idea remains important and I recommend publication.


Review 3

Summary and Contributions: This paper present an approach to modeling and forecasting multivariate time series using Gaussian state space models augmented with normalizing flow-based emission distributions. This model combination results in efficient closed-form filtering, smoothing and forecasting, whereas competing approaches based on other types of emission distributions require significantly more complex approximate inference.

Strengths: * Flexible modeling and forecasting multivariate time series is an important and relevant problem for the NeurIPS community. * The presented model combines well established components (Gaussian state space models and normalizing flows), but this specific combination using invertible emission transformations results in very efficient inference while accommodating multiple dimensional data as well as non-linearity. This is quite significant.

Weaknesses: * The fact that the approach can deal with missing data in the sense of completely missing multivariate observations is useful. However, the paper does not consider the case where a multivariate vector of observations at a given time point is only partially observed. This is also a very common case (perhaps more so than vectors being completely missing). The authors should comment on whether their approach can be adapted to cover this case and if so how. * The description of the qualitative experiments in Section 4.1 is overly terse. The graphs are missing axis labels and are also difficult to interpret. Also representing the data in the time domain would be helpful.

Correctness: The approach appears to be technically correct. It is unclear whether there is any representational capacity difference between having fully independent latent time series or not when normalizing flows are used to mix across latent time series. Please comment on this point. If there is no representational capacity difference, it would be preferable to present the simpler independent version of the model as the canonical version (similar to factor analysis and VAEs). The empirical methodology used to evaluate methods appears to be correct in terms of what is stated. However, the methodology used to select hyper-parameters for the real data experiments is not mentioned at all in the main paper (only a train/test split is described). The supplemental material mentions "We use the same hyperparameters for the model architecture across all datasets." Please provide specific details on which hyper-parameters were optimized and how for the proposed approach and the other approaches compared to.

Clarity: The paper is well written in general. As noted above, the clarity of the presentation of the qualitative experimental results could be improved. There are also some issues, like with missing details about hyper-parameter optimization, that need to be dealt with in the main paper for improved clarity.

Relation to Prior Work: This work is very close to a number of alternative approaches that extend latent linear Gaussian models and variational autoencoders in different ways. However, I believe this specific combination has not been presented previously and it has significant advantages over competing approaches in this space.

Reproducibility: No

Additional Feedback: See comments above for specific questions to address in the author response. Post-Feedback Update: The authors have addressed all of my questions about the paper. I appreciate the detailed response regarding partially observed data. Adding this discussion to the paper would be helpful.

[Author Response · NeurIPS 2020]

We thank all the reviewers for their insightful and positive feedback. We are delighted they enjoyed reading the paper,
and they found our approach to be interesting and useful. We are also very pleased the reviewers perceived the approach
to have significant potential, beyond just performance gains.

**Common comments:** We agree that the clarity of qualitative experiments section could be improved by giving a better
description of Figure 2 (R3) and adding missing axis labels (R5). The detailed description of these experiments had to
be moved to the Appendix due to space limitations. We will make this section clearer in the final version.

We will now address the reviewers' comments individually.

**Reviewer 3**:
i) We will add discussion about the relation between our method and Kernel Kalman Filters, State Space Gaussian
Process (GP) approach as well as GP latent variable model in the final version. We also note that GP-Copula [1] used in
the evaluation is an example of non-Gaussian AR methods.
ii) *hyperparams ('observation' and 'state' noise variances) are sampled:* It is not the variances that are sampled
but the random noise in order to sample from the forecast distribution. The observation and state variances are not
hyperparameters but are actually predicted from the RNN, using the associated covariates. Here sampling is done
because in applications it is preferable to represent the forecast distribution in the form of Monte Carlo samples [1, 8, 9].
iii) *The missing samples approach seems the same as that proposed for the KF by Shumway and Stoffer:* Thank you for
the reference. We agree it is a standard procedure and will add a citation.

**Reviewer 4**:
i) *Placing the emission noise before the nonlinearity is a crucial move; omission of any discussion of the non-additive*
*noise:* You are perfectly right that placing the noise before the non-linearity is crucial to obtain tractability for filtering.
However, this does not imply that data generated from a process where additive noise is added after the non-linear
function cannot be modelled: in fact we use this particular model (nonlinear transformation with additive non-Gaussian
noise) to generate data and test our method in the qualitative experiments in Section 4.1 (refer to appendix C.1 for
details). We will make this more explicit in the final version.
ii) *This technique can be applied as a drop-in replacement in many models:* Yes, we believe so.
iii) *Simple example:* There may be issues if the normalizing flow is highly skewed, however we have not encountered
this issue in practice.
iv) *Qualitative experiments seemed particularly artificial:* While the main purpose of these experiments is a sanity check
for the overall method, the artificial data generated still has all the intricacies of real world scenarios like non-Gaussian
*time varying* observation noise coupled with a seasonal behaviour and non-linear dependencies between the time series.
The complex nature of the time series observations are better highlighted in appendix Section C.1.2 when plotted
against time. Also note that the generative model used to simulate artificial data is not exactly same as our model but
instead uses non-Gaussian additive noise after nonlinear transformation of the state.
v) *Results of GP-Copula for some datasets differ between your paper and [1]:* The results in the paper of GP-Copula
use a flawed variant of CRPS-Sum *without* normalizing the time-series beforehand as in the data, the time-series have
drastically different scales, hence not properly evaluating the captured dependencies across time-series.
vi) Thanks for the detailed comments in the additional feedback section. We will incorporate these in the final version.

**Reviewer 5**:
i) *Partial observations:* The local-global instantiation presented in Section 3 could deal with partial observations, but
would require marginalising over the missing dimensions resorting to a Monte Carlo approximation. Alternatively,
if one is interested in dealing with partial observations, it is possible to consider an instantiation with a global LGM
with non-diagonal covariance matrix modelling dependencies across time-series, and a normalising flow is applied
locally to each time-series (as in our ablation study model $f_t$ Local). In this case, the marginalisation of any missing set
of time-series actually yields an analytic form for the filtering, smoothing and forecast distributions, and handling of
partial observations can be efficiently dealt with. Note that many nonlinear methods do not achieve this. We will be
happy to respond to this in the final version.
ii) *Representational capacity difference between having fully independent latent time series or not when normalizing*
*flows are used to mix across latent time series?* Just as you have said, we believe that without loss of expressivity,
the latent time-series can be considered independent, assuming that the normalizing flow itself is expressive enough.
However it may be still be useful to consider the model in its general form for completeness, e.g. as one may have some
prior knowledge in the form of the latent LGM that one may wish to encode explicitly.
iii) *Hyper-parameter optimization in the main paper:* We agree that although we mention that we use a validation set in
the appendix Section C, this should also be contained in the main paper. This will be modified. In order to select the
hyper-parameters, we use a validation set, which is derived from the training set: we cut the original training set in time
into two parts, taking the most recent part for validation, so as to have a validation set of the same size as the test set.

[Meta-Review · NeurIPS 2020]

We thank you for your submission. Reviewers agree that the paper is novel and of interest to the Neurips community. Please address carefully reviewers comments in your camera ready version.